# Efficacy of Lenvatinib and Sorafenib in the Real-World First-Line Treatment of Advanced-Stage Hepatocellular Carcinoma in a Taiwanese Population

**DOI:** 10.3390/jcm11051444

**Published:** 2022-03-06

**Authors:** Shou-Wu Lee, Sheng-Shun Yang, Han-Chung Lien, Yen-Chun Peng, Chung-Wang Ko, Teng-Yu Lee

**Affiliations:** 1Division of Gastroenterology, Department of Internal Medicine, Taichung Veterans General Hospital, Taichung 40705, Taiwan; ericest@vghtc.gov.tw (S.-W.L.); eric@vghtc.gov.tw (S.-S.Y.); ericest429@gmail.com (H.-C.L.); ericest324134@gamil.com (Y.-C.P.); eric324134@gmail.com (C.-W.K.); 2Department of Internal Medicine, Chung Shan Medical University, Taichung 40201, Taiwan; 3Department of Internal Medicine, Yang Ming Chiao Tung University, Taipei 112304, Taiwan; 4Department of Post-Baccalaureate Medicine, College of Medicine, Chung Hsing University, Taichung 40227, Taiwan; 5Ph.D. Program in Translational Medicine, Chung Hsing University, Taichung 40227, Taiwan; 6Institute of Biomedical Sciences, Chung Hsing University, Taichung 40227, Taiwan

**Keywords:** hepatocellular carcinoma, lenvatinib, sorafenib

## Abstract

Aim: Currently, atezolizumab combined with bevacizumab is the standard first-line treatment for unresectable hepatocellular carcinoma (HCC), but lenvatinib or sorafenib are still recommended for these patients for some reasons. The aim of the study was to determine the outcomes of Taiwanese patients with advanced-stage HCC who received lenvatinib or sorafenib. Methods: Data on patients with BCLC stage C HCC who were receiving lenvatinib or sorafenib as the first-line therapy from May 2018 to August 2020 was collected. The individuals with lenvatinib and sorafenib were propensity score-matched at a ratio of 1:2. Results: A total of 22 patients with lenvatinib and 44 patients with sorafenib were enrolled. The ORR (36.4% vs. 11.4%, *p* = 0.023) and DCR (81.9% vs. 56.9%, *p* = 0.039) were both higher in the lenvatinib group compared with the sorafenib group. The median overall survival (OS) of the lenvatinib group and the sorafenib group was 9.36 months and 8.36 months, respectively. The best median OS was detected in patients receiving lenvatinib and having an objective tumor response (11.29 months), with a significant difference (*p* = 0.031) compared with the other groups. Conclusion: Lenvatinib, compared to sorafenib, had better ORR and DCR, but similar OS, in Taiwanese patients with advanced-stage HCC. The patients with an objective tumor response had a better OS.

## 1. Background

Hepatocellular carcinoma (HCC) is the most frequent primary liver cancer, and the incidence is expected to increase as a consequence of chronic liver disease, including chronic hepatitis B virus (HBV) and hepatitis C virus (HCV) infections and excessive alcohol consumption [1]. Treatments of HCC depend on disease stages, typically according to the Barcelona Clinic Liver Cancer (BCLC) stage, which considers prognosis-related factors such as tumor burden, liver function, and performance status [2]. In patients with advanced (BCLC stage C) HCC, systemic therapy with tyrosine kinase inhibitor (TKI) or immunotherapeutic agents are used with the aim of prolonging survival by slowing tumor progression [3,4].

Sorafenib has been the standard first-line treatment for unresectable HCC since 2007, when the SHARP trial demonstrated that sorafenib improved median overall survival (OS) compared to placebo in patients who had not received prior systemic therapy (10.7 vs. 7.9 months, HR = 0.69, *p* < 0.001) [3]. The subsequent Asia-Pacific trial confirmed these results in Asian patients (6.5 vs. 4.2 months, HR = 0.68, *p* = 0.014) [5]. In 2018, lenvatinib became a new choice that could be added to the regimen for unresectable HCC, based on the REFLECT Phase III trial, which showed noninferior median OS compared to sorafenib (13.6 vs. 12.3 months, HR = 0.92) [6].

More recently, atezolizumab combined with bevacizumab resulted in better OS and progression-free survival (PFS) than sorafenib in patients with unresectable HCC (12-month OS, 67.2% vs. 54.6%; median PFS, 6.8 vs. 4.3 months, HR = 0.59, *p* < 0.001) [7]. The regimen currently replaces the recommended first-line therapeutic regimen for unresectable HCC. However, due to reasons such as health insurance coverage, high cost, and the unavailability of new drugs, lenvatinib or sorafenib remain the first-line therapeutic regimens for these patients in Taiwan.

The real-world data on the therapeutic benefits observed after lenvatinib compared to sorafenib are still limited in the Taiwanese population. The aim of the present study was to determine the outcomes of Taiwanese patients with advanced-stage HCC who received lenvatinib or sorafenib as the first-line therapy.

## 2. Methods

Data for subjects with HCC, BCLC stage C, as diagnosed according to the American Association for the Study of Liver Disease (AASLD) guidelines [8], and who were receiving TKI as the first-line monotherapy from May 2018 to August 2020 were retrospectively collected and evaluated. All data were fully anonymized before assessment. All enrolled cases were categorized as Child–Pugh class A and no previous exposure to HCC systemic therapies, such as TKI or immunotherapeutic agents. The general data of the enrolled patients, including age, gender, presence of chronic HBV, HCV infection, macroscopic vascular invasion (MVI), or extrahepatic spread (EHS), along with their laboratory data, including the serum level of bilirubin, alanine aminotransferase (ALT), and alpha-fetoprotein (AFP), were recorded for each individual. The exclusion criteria included cases diagnosed with Child–Pugh stage B or C, BCLC stage A or B HCC, poor performance status, a lack of compliance to drugs, survival of less than two months, the absence of radiologic examination, or the combined treatment of immunotherapeutic agents within the following day.

The individuals with lenvatinib were identified and they were then propensity score-matched at a ratio of 1:2 with cases treated using sorafenib. After administering lenvatinib or sorafenib, the subjects were followed up in the outpatient clinic every 2 to 4 weeks. The duration of TKI usage was determined by each patient’s hepatologist, but usually discontinued once obvious tumor progression was disclosed by subsequent imaging studies. The therapeutic duration of TKI for each enrolled subject was recorded.

Tumor response on images was assessed every 4 to 8 weeks by 5 fixed radiologists, who have excellent experience of over 10 years in this field. The assessment of the best tumor response was done according to the modified RECIST (mRECIST) criteria [9] with four response categories: complete response (CR), partial response (PR), stable disease (SD), and progressive disease (PD). The patients with CR or PR were categorized into the objective response (OR) group.

Adverse events (AEs) were defined as the appearance of hand–foot skin reaction (HFSR), hypertension, diarrhea, or fatigue after the administration of TKI. The associations between the clinical parameters and the efficacy of TKI were analyzed. Overall survival (OS) was defined as the time from the start of TKI until death or the last follow-up and presented as a median value with a 95% confidence interval (CI).

Data are expressed as the standard deviation of the mean for each of the measured parameters. The positive rates of each stratified group are expressed as a percentage of the total patient number. Statistical comparisons were made using Pearson’s Chi-square test in order to compare the effects of the positive rate of each stratified group. An independent *t*-test was used to analyze continuous variables. A *p*-value below 0.05 was considered statistically significant. Survival analysis was carried out using the Kaplan–Meier method for univariate analysis and comparisons were subsequently performed with the log-rank test.

## 3. Results

Initially, there were 47 patients with lenvatinib and 102 patients with sorafenib. After excluding by exclusion criteria and propensity score-matched, a total of 22 patients with lenvatinib and 44 patients with sorafenib were enrolled, as displayed in Figure 1, and the characteristics of these cases are shown in Table 1. The median age of the two groups was from 63.95 to 63.77 years, and male predominance (81.8%) was noted. Prevalence of chronic HBV and HCV infection was 54.5% and from 27.3% to 29.5%, respectively. All patients belonged to Child–Pugh class A and BCLC stage C. Thirteen cases (59.1%) and eleven cases (50.0%) in the lenvatinib group, and twenty-five cases (56.8%) and twenty-three cases (52.3%) in the sorafenib group, had MVI and EHS, respectively. The laboratory parameters, including total bilirubin and ALT in the two groups, were similar. The average of the AFP of the lenvatinib group was nonsignificantly higher than that of the sorafenib group (mean 6.17 vs. 2.53 × 104 ng/mL, *p* = 0.214). Furthermore, a significantly higher proportion of patients treated with lenvatinib had an AFP ≧ 400 ng/mL compared with those receiving sorafenib (72.7% vs. 40.9%, *p* = 0.019). The therapeutic durations of the lenvatinib group and the sorafenib group were 4.59 ± 1.91 and 5.22 ± 3.76 months, respectively.

The outcomes of the enrolled patients who underwent TKI are listed in Table 2. The numbers of cases with PR, SD, and PD were eight (36.4%), ten (45.5%), and four (18.1%) in the lenvatinib group, and five (11.4%), 20 (45.5%) and nineteen (43.1%) in the sorafenib group, respectively. Overall, the objective response rate (ORR) was 36.4% and 11.4% in the lenvatinib group and the sorafenib group, respectively. The disease control rate (DCR) was 81.9% and 56.9% in the lenvatinib group and the sorafenib group, respectively, both of which were significant (*p* = 0.023 and 0.039).

The AEs detected in each group are shown in Table 3. All AEs belonged from grade 1 to 2. No patients needed to discontinue treatment, and the dosage reduction occurred in four (18.2%) and six (13.6%) of sorafenib and lenvatinib group, respectively. Patients with lenvatinib had a higher prevalence rate of hypertension (27.3% vs. 13.6%) and diarrhea (27.3% vs. 15.9%) compared to those treated with sorafenib. In contrast, the sorafenib group had a higher prevalence of HFSR (50.0% vs. 31.8%) and fatigue (31.8% vs. 27.3%) than the lenvatinib group. However, no significant differences existed. 

As shown in Figure 2, the median OS (95% CI) of the lenvatinib group and the sorafenib group was 9.36 months (7.69–11.04) and 8.36 months (6.93–9.79), respectively, and the difference was nonsignificant (*p* = 0.107). Further analysis of the OS stratified by tumor radiological response in each group is shown in Figure 3. In the lenvatinib group, the median OS (95% CI) was 11.29 months (9.17–13.42) and 8.41 months (6.27–10.56) in patients with and without the OR, respectively. In the sorafenib group, the median OS (95% CI) was 10.60 months (3.57–17.63) and 8.08 months (6.71–9.44) in the cases with and without OR, respectively. Significant differences existed (*p* = 0.031) between the subgroups of patients receiving lenvatinib with OR and the other subgroups.

## 4. Discussion

The incidence of HCC has been steadily rising, and most patients with HCC present with an intermediate (BCLC stage B) or advanced stage (BCLC stage C) when curative therapies are no longer possible. Treatment recommendations for such subjects include locoregional therapy for BCLC stage B HCC, such as TACE, and systemic therapy for BCLC stage C, such as TKI or immunotherapeutic agents [10]. Currently, lenvatinib and sorafenib are considered to be the standard method of care for BCLC stage B HCC [3,5,6].

Sorafenib is a TKI targeting the RAF/MEK/ERK axis of the RAS cascade signal, vascular endothelial growth factor receptors (VEGFRs) 1–3, and the platelet-derived growth factor receptor β (PDGFR-β) [11]. Sorafenib is the first-line systemic therapy approved for the treatment of unresectable HCC based on the results of the multicenter, randomized, phase III SHARP trial and the Asia-Pacific trial [4,5].

Lenvatinib is a TKI targeting VEGFR 1–3, PDGFR-α, fibroblast growth factor receptor 1–4 (FGFR 1–4), KIT, and rearrangement during transfection (RET) [12]. According to the pharmacokinetic (PK) analysis, including patients with HCC and Child–Pugh class A, the optimal dose according to body weight is 12 mg once daily for patients ≧60 kg and 8 mg once daily for patients <60 kg [13].

Lenvatinib has been approved as the first-line treatment of unresectable HCC based on the open-label, noninferiority phase III REFLECT trial [6]. The REFLECT trial enrolled HCC patients untreated with systemic therapy, with BCLC stage B or C, preserved liver function (Child–Pugh class A), and a good performance status. In the finial part of the study, 954 patients were randomized to receive lenvatinib (N = 478) or sorafenib (N = 476) between March 2013 and July 2015. Baseline patient characteristics were well balanced between the two treatment groups, except for a higher rate of HCV infection and lower AFP baseline levels in the sorafenib group. The ORR according to the mRECIST criteria was higher in the lenvatinib arm compared to the sorafenib arm (24.1% vs. 9.2%, *p* < 0.001). Median treatment duration was 5.7 months with lenvatinib and 3.7 months with sorafenib. Median OS was 13.6 months on lenvatinib and 12.3 months on sorafenib, with an HR of 0.92 (95% CI = 0.79–1.06), demonstrating the noninferiority of lenvatinib compared to sorafenib. In the lenvatinib arm, the most common AEs were hypertension (42%), diarrhea (39%), anorexia (34%), weight loss (31%), and proteinuria (25%) [6].

A retrospective three-center real-world study on 92 patients in Korea reported an ORR of 21.1% and a lower median OS for patients with Child–Pugh class B (B vs. A, 5.3 months vs. 10.7 months) [14]. Similarly, a retrospective real-world multicenter analysis of 181 patients in Japan found Child–Pugh class A (A vs. B, *p* = 0.007) and BCLC stage B (B vs. C, *p* = 0.002) were associated with better OS following lenvatinib treatment. Moreover, the ORR was also significantly higher in the Child–Pugh subclass A5 (44%) compared with the other subclasses [15]. In another real-world study, data from a Canadian multicenter database which enrolled 220 patients found the ORR and median OS were 22% and 13 months, respectively, and the outcomes were similar between lenvatinib as the first-line and as the late-line therapeutic regimen of HCC [16]. Recent real-world data including 466 patients in Italy reported the median PFS as being 9.0 and 4.9 months for the lenvatinib and sorafenib arm, respectively. Patients treated with lenvatinib showed a higher percentage of response rate (29.4% vs. 2.8%; *p* < 0.00001) compared with those treated with sorafenib [17].

Our study was designed as a retrospective comparison of lenvatinib and sorafenib for the first-line therapeutic agent of unresectable HCC. Our enrolled patients all belonged to BCLC stage C, Child–Pugh class A, and were matched by propensity score between the lenvatinib group and the sorafenib group. The ORRs were 36.4% and 11.4% in the lenvatinib group and the sorafenib group, respectively. The median OS was 9.36 months and 8.36 months in the lenvatinib arm and the sorafenib arm, respectively. The poor OS of our results compared with the REFLECT trial might be due to the fact that some subjects in our study had extensive liver involvement or MVI, whereas such cases were excluded in the REFLECT trial [6]. Our results reported the real-world data comparing the efficacy and safety between lenvatinib and sorafenib to the unresectable HCC in a Taiwanese population. Our results found patients with lenvatinib had a higher rate of hypertension (27.3% vs. 13.6%) and diarrhea (27.3% vs. 15.9%), and lower rates of HFSR (31.8% vs. 50.0%) and fatigue (27.3% vs. 31.8%), when compared to those with sorafenib. The lower prevalence of AEs in our study compared with the REFLECT trial [6] could be explained by the exclusion of patients with a lack of compliance and self-reported AEs, which possibly results in some AEs being underestimated. 

Not surprisingly, our results found that patients with OR had a better median OS than those without, both in the lenvatinib group and the sorafenib group. The individuals with OR in the lenvatinib arm had the best median OS (11.29 months) compared with the other subgroups, and the difference was significant (*p* = 0.031). 

There were several limitations in our study. First, this study was retrospective in nature and was conducted at a single tertiary care center. Selection bias may therefore have existed. Second, only subjects diagnosed with Child–Pugh class A and BCLC stage C HCC were enrolled in our study. Third, our sample size was relatively small and the follow-up period was relatively short. Lastly, we had no sufficient data of the regimen of atezolizumab/bevacizumab, which are recommended as more effective first-line therapeutic choices for unresectable HCC [7]. Further prospective research involving the analysis of more variables is therefore warranted.

## 5. Conclusions

Lenvatinib, compared to sorafenib, had a better ORR and DCR, but similar OS, in Taiwanese patients with advanced-stage HCC. Patients with objective tumor response had a significantly better overall survival.

## Figures and Tables

**Figure 1 jcm-11-01444-f001:**
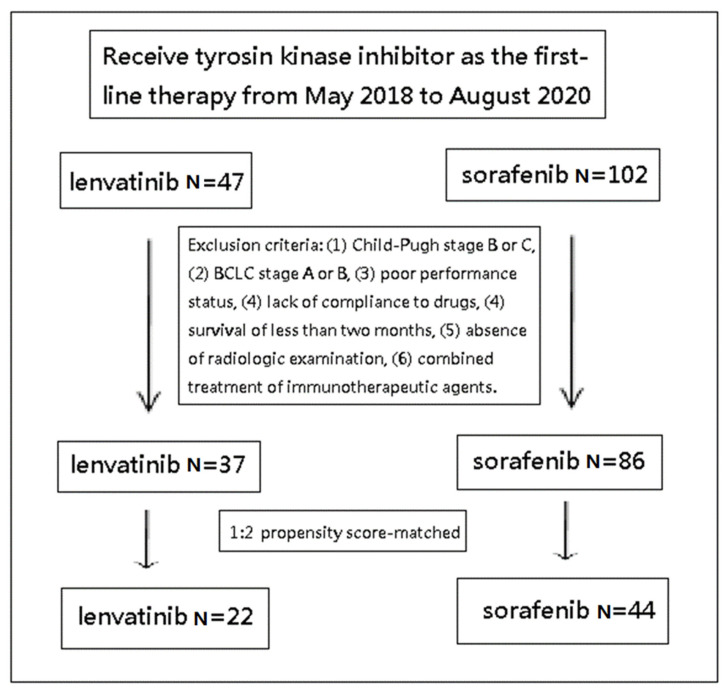
The overall survival of advanced-stage hepatocellular carcinoma patients receiving lenvatinib or sorafenib.

**Figure 2 jcm-11-01444-f002:**
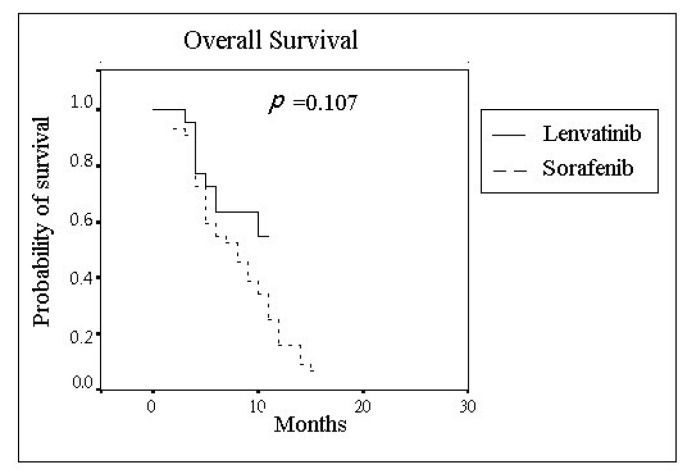
The overall survival of advanced-stage hepatocellular carcinoma patients receiving lenvatinib or sorafenib.

**Figure 3 jcm-11-01444-f003:**
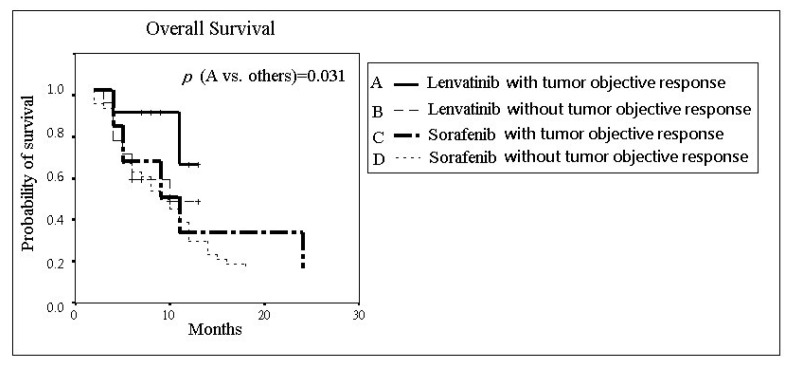
The overall survival of advanced-stage hepatocellular carcinoma patients receiving lenvatinib or sorafenib with or without tumor objective response.

**Table 1 jcm-11-01444-t001:** The general data of the patients with lenvatinib and the patients with sorafenib.

		Lenvatinib (N = 22)			Sorafenib (N = 44)			*p*-Value
		M ± SD	N	%	M ± SD	N	%	
Age		63.95 ± 11.38			63.77 ± 10.53			0.949 ^a^
Gender (male)			18	(81.8%)		36	(81.8%)	1.000 ^b^
Hepatitis infection	HBV		12	(54.5%)		24	(54.5%)	0.848 ^b^
	HCV		6	(27.3%)		13	(29.5%)	
Child–Pugh score	A		44	(100%)		90	(100%)	1.000 ^b^
BCLC stage	C		44	(100%)		90	(100%)	1.000 ^b^
MVI			13	(59.1%)		25	(56.8%)	0.860 ^b^
EHS			11	(50.0%)		23	(52.3%)	0.862 ^b^
Bilirubin (U/L)		0.79 ± 0.43			0.89 ± 0.46			0.394 ^a^
ALT (U/L)		68.91 ± 72.42			68.11 ± 71.91			0.953 ^a^
AFP (×10^4^ ng/mL)		6.17 ± 13.83			2.53 ± 9.46			0.214 ^a^
AFP (≧400 ng/mL)			16	(72.7%)		18	(40.9%)	0.019 ^b^
TKI therapeutic duration (months)		4.59 ± 1.91			5.22 ± 3.76			0.459 ^a^

The *p*-values were analyzed with an independent *t*-test ^a^; Pearson’s Chi-square test ^b^. Abbreviations: AFP, alpha-fetoprotein; ALT, alanine aminotransferase; EHS, extrahepatic spread; HBV, hepatitis B; HCV, hepatitis C; M, mean; MVI, macroscopic vascular invasion; N, number of patients; SD, standard derivation; TKI, tyrosine kinase inhibitor.

**Table 2 jcm-11-01444-t002:** The radiological tumor responses of the patients with lenvatinib and the patients with sorafenib.

	Radiological Best Overall Response				
	Lenvatinib (N = 22)		Sorafenib (N = 44)		*p*-Value
	N	%	N	%	
mRECIST					
Complete response	0		0		0.026
Partial response	8	(36.4%)	5	(11.4%)	
Stable disease	10	(45.5%)	20	(45.5%)	
Progressive disease	4	(18.1%)	19	(43.1%)	
ORR	8	(36.4%)	5	(11.4%)	0.023
DCR	18	(81.9%)	25	(56.9%)	0.039

All *p*-values were analyzed with Pearson’s Chi-square test. Abbreviations: DCR, disease control rate; N, number of patients; ORR, objective response rate.

**Table 3 jcm-11-01444-t003:** The adverse events of the patients with lenvatinib and the patients with sorafenib.

	Lenvatinib (N = 22)		Sorafenib (N = 44)		
	N	%	N	%	*p*-Value
HFSR	7	(31.8%)	22	(50.0%)	0.161
Hypertension	6	(27.3%)	6	(13.6%)	0.176
Diarrhea	6	(27.3%)	7	(15.9%)	0.274
Fatigue	6	(27.3%)	14	(31.8%)	0.705

All *p*-values were analyzed with Pearson’s Chi-square test. Abbreviations: HFSR, hand–foot syndrome reaction; N, number of patients.

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
