# Peer review of "Efficacy of Lenvatinib and Sorafenib in the Real-World First-Line Treatment of Advanced-Stage Hepatocellular Carcinoma in a Taiwanese Population"

_jcm, 2022, doi:10.3390/jcm11051444_

Round 1

Reviewer 1 Report

The paper by Lee et al. aimed to retrospectively compare the outcomes of sorafenib and lenvatinib treatment advanced HCC patients. Considering the expansion of systemic therapies alternatives, this is a relevant topic. However, the study has some major limitations.

1) This is a single center study including a very limited number of patients. Several other studies compared sorafenib and lenvatinib treatment in real life clinical setting (some studies have been cited by the Authors but also others have been published, such as doi: 10.2147/CMAR.S330195). What are the advantages of this study compared to previous literature and where does its novelty lie?

2) I have also some concerns about methodology, because it is not clear how the patients included in the analysis were selected. How many patients were treated with TKIs overall? The Authors reported some exclusion criteria, but it remains uncertain whether these patients were excluded from the treatment with TKIs or whether they were treated with TKIs but excluded from the analysis. In addition, the Authors claimed that patients were matched for age, sex and etiology, but it is not clear how were these patients matched (PSM?).

Being this study retrospective and being its aim to compare the efficacy of the two treatment schemes in real life clinical practice, I would suggest to compare the efficacy and the safety of the two drugs considering all the patients treated. Subsequently, a propensity score matching could be used to create two groups comparable for baseline characteristics.

3) Why the Authors did not consider and compare progression-free survival?

4) The Authors reported the number of adverse events during sorafenib and lenvatinib treatment. However, the grade of these adverse events, how they were managed and the consequences were not reported (Discontinuation of the treatment? Reduction in dosage?). None of the patients experienced other adverse events other than HFSR, hypertension, diarrhea and fatigue?

5) There are some typos that need to be corrected.

Author Response

The paper by Lee et al. aimed to retrospectively compare the outcomes of sorafenib and lenvatinib treatment advanced HCC patients. Considering the expansion of systemic therapies alternatives, this is a relevant topic. However, the study has some major limitations.

1) This is a single center study including a very limited number of patients. Several other studies compared sorafenib and lenvatinib treatment in real life clinical setting (some studies have been cited by the Authors but also others have been published, such as doi: 10.2147/CMAR.S330195). What are the advantages of this study compared to previous literature and where does its novelty lie?

Answer: Thanks for your commend. Our results reported the real-world data comparing the efficacy and safety between lenvatinib and sorafenib to the unresctable HCC in a Taiwanese population, which is different with previous studies. We have added this points to the Topic, abstract, and discussion section. Besides, we also added the article (doi: 10.2147/CMAR.S330195) into our reference.

2) I have also some concerns about methodology, because it is not clear how the patients included in the analysis were selected. How many patients were treated with TKIs overall? The Authors reported some exclusion criteria, but it remains uncertain whether these patients were excluded from the treatment with TKIs or whether they were treated with TKIs but excluded from the analysis. In addition, the Authors claimed that patients were matched for age, sex and etiology, but it is not clear how were these patients matched (PSM?). Being this study retrospective and being its aim to compare the efficacy of the two treatment schemes in real life clinical practice, I would suggest to compare the efficacy and the safety of the two drugs considering all the patients treated. Subsequently, a propensity score matching could be used to create two groups comparable for baseline characteristics.

Answer: Thanks for your commends. As the statement of Methods section, our enrolled patents are receiving TKI as the "first-line" monotherapy to advanced-stage HCC. The cases exposured to previous TKI or IO treatment are excluded. We have added this point to the Methods section. Besides, after re-analyze our enrolled cases, the propensity score between lenvatinib and sorafenib are similar. Thus, we have corrected the statements in Abstract and Method sections.

3) Why the Authors did not consider and compare progression-free survival?

Answer: Thanks for your commends. Indeed, the progression-free survival is a important outcome in clinical study. However, to the oncology field, overall survival should be the most important and final decision to determine the drug efficacy to each cancer. Therefore, we compared the overall survival, rather than progression-free survival, in our study.

4) The Authors reported the number of adverse events during sorafenib and lenvatinib treatment. However, the grade of these adverse events, how they were managed and the consequences were not reported (Discontinuation of the treatment? Reduction in dosage?). None of the patients experienced other adverse events other than HFSR, hypertension, diarrhea and fatigue?

Answer: Thanks for your commend. Because our exclusion criteria include lack of compliance to drugs, our enrolled patients are tolerable to sorafenib or lenvatinib. All adverse events belonged to grade 1 to 2.No patients need to discontinue treatment, and dosage reduction occurred in 4 (18.2%) and 6 (13.6%) of sorafenib and lenvatinin group. We have added these statements to the Result section.

Reviewer 2 Report

-First-line systemic treatment of advanced HCC is still an unresolved issue. This study offers an interesting comparison in the clinical practice scenario. However, it would be recalled that several prognostic scores have been proposed to better assist sorafenib treatment and select HCC patients having high chance of treatment response as recently reported (Comparison of Prognostic Scores in Patients With Hepatocellular Carcinoma Treated With Sorafenib. Clin Transl Gastroenterol. 2021 Jan 14;12(1):e00286).

-A clinically relevant point is the sequential treatment after first-line failure. According to current international guidelines, second-line systemic treatments after first-line failure have been established only after sorafenib failure, and seuqntial treatment with sorafenib-regorafenib provide significant overall survival gain as recently reported also in clinical practice studies (Experience with regorafenib in the treatment of hepatocellular carcinoma. Therap Adv Gastroenterol. 2021 May 28;14:17562848211016959).

-The authors should also discuss the impact of previous HCC treatments, in particular transarterial chemioembolization. During the execution of the TACE procedures there is an amount of absorbed radiant dose, for operators (interventional radiologists) and for Patients. A correct standardization of the procedures, it is necessary to reduce the number of rays absorbed for both patients and radiologists in order to reduce the possible stochastic and no-stochastic effects [Med Phys. 2012;39(5):2491-2498. doi:10.1118/1.3702457]. In my opinion, the interventional radiologists who performed TACE should be necessary expert radiologists in order to reduce the dose in patients that received these treatments [Med Phys. 2012;39(5):2491-2498. doi:10.1118/1.3702457]. According to the aforementioned paper, these treatments need to be performed by expert interventional radiologists, to reduce the risks of x-ray exposure, reducing its feasibility. According to the Authors opinion, this problem could open new scenarios for systemic therapies? Please, discuss this theme.

- In my opinion, the assessment of treatment responses using mRECIST should be expanded with comments regarding mRECIST. This point is very important, and the authors have to discuss it. It is well known that the decision on whether an HCC patient is a responder or progressor after treatment may vary among different radiologists, especially in case of a non-specifically trained radiologist and, therefore, regardless of the adopted criteria, patients should be evaluated by experienced radiologists to minimize variability in this critical instance [Eur Radiol. 2018;28(9):3611-3620. doi:10.1007/s00330-018-5393-3]. Please, could the Authors report that it is important that experienced radiologists evaluate imaging, and that one or more experienced radiologists must evaluated imaging according to mRECIST. Please, cite the aforementioned paper [Eur Radiol. 2018;28(9):3611-3620. doi:10.1007/s00330-018-5393-3] and report if the radiologists were expert (how many years?).

Author Response

First-line systemic treatment of advanced HCC is still an unresolved issue. This study offers an interesting comparison in the clinical practice scenario. However, it would be recalled that several prognostic scores have been proposed to better assist sorafenib treatment and select HCC patients having high chance of treatment response as recently reported (Comparison of Prognostic Scores in Patients With Hepatocellular Carcinoma Treated With Sorafenib. Clin Transl Gastroenterol. 2021 Jan 14;12(1):e00286).

Answer: Thanks for your commend. Indeed, the prognostic scores to predict the efficacy of TKI or IO to advanced HCC could be a guild to select patients with high chance of treatment response. As the article statements (Clin Transl Gastroenterol. 2021 Jan 14;12(1):e00286), the factors of prognostic scores include albumin, bilirubin, presence of major vessel encasement or extrahepatic spread. In our study, all the enrolled patients belong to Child-Pugh A and BCLC stage C, and our reports focus on the comparison of the efficacy and safety between lenvatinib and sorafenib treatments.

-A clinically relevant point is the sequential treatment after first-line failure. According to current international guidelines, second-line systemic treatments after first-line failure have been established only after sorafenib failure, and seuqntial treatment with sorafenib-regorafenib provide significant overall survival gain as recently reported also in clinical practice studies (Experience with regorafenib in the treatment of hepatocellular carcinoma. Therap Adv Gastroenterol. 2021 May 28;14:17562848211016959).

Answer: Thanks for your commend. Indeed, the second-line systemic treatments are important to the HCC patients failed to first-line treatment. As the statements in the Topic and Method section, our enrolled cases received first-line treatment to HCC, and our reports focus on the comparison of the efficacy and safety between lenvatinib and sorafenib treatments.

-The authors should also discuss the impact of previous HCC treatments, in particular transarterial chemioembolization. During the execution of the TACE procedures there is an amount of absorbed radiant dose, for operators (interventional radiologists) and for Patients. A correct standardization of the procedures, it is necessary to reduce the number of rays absorbed for both patients and radiologists in order to reduce the possible stochastic and no-stochastic effects [Med Phys. 2012;39(5):2491-2498. doi:10.1118/1.3702457]. In my opinion, the interventional radiologists who performed TACE should be necessary expert radiologists in order to reduce the dose in patients that received these treatments [Med Phys. 2012;39(5):2491-2498. doi:10.1118/1.3702457]. According to the aforementioned paper, these treatments need to be performed by expert interventional radiologists, to reduce the risks of x-ray exposure, reducing its feasibility. According to the Authors opinion, this problem could open new scenarios for systemic therapies? Please, discuss this theme.

Answer: Thanks for your commend. Indeed, previous TACE status might influence the later outcomes with systemic treatments, and it may need more discussion. In our study, all the enrolled patients received first-line treatment to advanced HCCs, and no previous other treatments could be detected, including TACE. Therefore, we have not added associated discussion in our manuscript.

- In my opinion, the assessment of treatment responses using mRECIST should be expanded with comments regarding mRECIST. This point is very important, and the authors have to discuss it. It is well known that the decision on whether an HCC patient is a responder or progressor after treatment may vary among different radiologists, especially in case of a non-specifically trained radiologist and, therefore, regardless of the adopted criteria, patients should be evaluated by experienced radiologists to minimize variability in this critical instance [Eur Radiol. 2018;28(9):3611-3620. doi:10.1007/s00330-018-5393-3]. Please, could the Authors report that it is important that experienced radiologists evaluate imaging, and that one or more experienced radiologists must evaluated imaging according to mRECIST. Please, cite the aforementioned paper [Eur Radiol. 2018;28(9):3611-3620. doi:10.1007/s00330-018-5393-3] and report if the radiologists were expert (how many years?).

Answer: Thanks for your commend. The assessments of mRECIST in HCC treatment responses in our center are underwent by 5 fixed radiologists with excellent experiences in this field. The average times of these experienced radiologists in this field are about 10-20 years. We have put this point in the Method discussion.

Round 2

Reviewer 1 Report

The Authors answered to all my comments. However, I still think that a better description of methods should be provided. Indeed, I have no clear how patients were selected for this study.

How many patients were treated with TKIs (sorafenib or lenvatinib) during the study period (May 2018 - August 2020)? 

If all patients treated with TKIs in the study period, who fulfilled the inclusion characteristics ("subjects with HCC, BCLC stage C, as diagnosed according to the American Association for the Study of Liver Disease (AASLD) guidelines, and who were receiving TKI as the first-line monotherapy from May 2018 to August 2020") were 66 (n=22 lenvatinib and n=44 sorafenib), no mention to matching at all should be made because these patients were not actually matched, but had simply similar basal characteristics.

If patients treated with TKIs in the study period were more than those included in this study, this number should be included in the text. Were all sorafenib and lenvatinib treated patients comparable for baseline characteristics? If patients were comparable for baseline characteristics there is no need to create two groups matched with propensity score. On the contrary, if patients were not comparable (e.g., for variables such as sex, age and etiology, as the Authors initially included in the manuscript) it is reasonable to balance for these characteristics with a propensity score matching.

II believe that a better clarification of the process of patients selection should be made. Probably, a flow chart of patients selection could be helpful.

Author Response

How many patients were treated with TKIs (sorafenib or lenvatinib) during the study period (May 2018 - August 2020)? 

Answer: Thanks for your commend. There were 47 patients with lenvatinib and 102 patients with sorfaenib during May 2018-August 2020, and we have add this statement into the Result section.

If all patients treated with TKIs in the study period, who fulfilled the inclusion characteristics ("subjects with HCC, BCLC stage C, as diagnosed according to the American Association for the Study of Liver Disease (AASLD) guidelines, and who were receiving TKI as the first-line monotherapy from May 2018 to August 2020") were 66 (n=22 lenvatinib and n=44 sorafenib), no mention to matching at all should be made because these patients were not actually matched, but had simply similar basal characteristics. If patients treated with TKIs in the study period were more than those included in this study, this number should be included in the text. Were all sorafenib and lenvatinib treated patients comparable for baseline characteristics? If patients were comparable for baseline characteristics there is no need to create two groups matched with propensity score. On the contrary, if patients were not comparable (e.g., for variables such as sex, age and etiology, as the Authors initially included in the manuscript) it is reasonable to balance for these characteristics with a propensity score matching.I believe that a better clarification of the process of patients selection should be made. Probably, a flow chart of patients selection could be helpful.

Answer: Thanks for your commend. There were 47 patients with lenvatinib and 102 patients with sorfaenib during May 2018-August 2020, and we have add this statement into the Result section.
Answer: Thanks for your commend. The two groups of sorafenib and lenvatinib treated patients were comparable for baseline characteristics, but propensity score-matched study design was still adapted to avoid retrospective study bias. Besides, we have added a flow chart of patients selection as Fig1.
